# Robotized Cochlear Implantation under Fluoroscopy: A Preliminary Series

**DOI:** 10.3390/jcm12010211

**Published:** 2022-12-27

**Authors:** Thierry Mom, Mathilde Puechmaille, Mohamed El Yagoubi, Alexane Lère, Jens-Erik Petersen, Justine Bécaud, Nicolas Saroul, Laurent Gilain, Sonia Mirafzal, Pascal Chabrot

**Affiliations:** 1Department of Otolaryngology Head Neck Surgery, University Hospital Center, Hospital Gabriel Montpied, 58, Rue Montalembert, 63000 Clermont-Ferrand, France; 2Mixt Unit of Research (UMR) 1107, National Institute of Health and Medical Research (INSERM), University of Clermont Auvergne (UCA), 63000 Clermont-Ferrand, France; 3Department of Radiology, University Hospital Center, Hospital Gabriel Montpied, 58, Rue Montalembert, 63000 Clermont-Ferrand, France

**Keywords:** cochlear implant, electrode-array insertion, Robot, fluoroscopy, scala translocation

## Abstract

It is known that visual feedback by fluoroscopy can detect electrode array (EA) misrouting within the cochlea while robotized EA-insertion (rob-EAI) permits atraumatic cochlear implantation. We report here our unique experience of both fluoroscopy feedback and rob-EAI in cochlear implant surgery. We retrospectively analyzed a cohort of consecutive patients implanted from November 2021–October 2022 using rob-EAI, with the RobOtol^®^, to determine the quality of EA-insertion and the additional time required. Twenty-three patients (10 females, 61+/−19 yo) were tentatively implanted using robot assistance, with a rob-EAI speed < 1 mm/s. Only three cases required a successful revised insertion by hand. Under fluoroscopy (*n* = 11), it was possible to achieve a remote rob-EAI (*n* = 8), as the surgeon was outside the operative room, behind an anti-radiation screen. No scala translocation occurred. The additional operative time due to robot use was 18+/−7 min with about 4 min more for remote rob-EAI. Basal cochlear turn fibrosis precluded rob-EAI. In conclusion, Rob-EAI can be performed in almost all cases with a low risk of scala translocation, except in the case of partial cochlear obstruction such as fibrosis. Fluoroscopy also permits remote rob-EAI.

## 1. Introduction

Cochlear implantation is a treatment employed all over the world to rehabilitate hearing in the case of severe to profound deafness [1,2,3] The surgical procedure is well-codified and comprises the insertion of an electrode-array (EA) within the cochlear scala tympani, where the spiral ganglion of the cochlear nerve can be electrically stimulated by the cochlear implant. Great advances in cochlear implantation have been achieved in recent years, making it possible, for example, in selected cases, to preserve residual hearing [4,5,6]. It has been shown that cochlear implantation can be less invasive and deleterious to the cochlea, if a thin EA is inserted at low speed [7]. Although EA-translocation from the scala tympani into the scala vestibuli could result in damage to the fine cochlear structure, controversy exists in the literature, with some teams reporting better results in well-placed EAs within the scala tympani [8,9,10,11,12], while others did not [13,14]. Visual feedback through fluoroscopy has shown that it is possible to detect early in real -time errors such as EA-misrouting, EA-tip foldover, or EA-tip blockage within the cochlea and to immediately correct them [15]. Recently, a robotized EA insertion (rob-EAI) technique has been shown to perform less traumatic insertion, with a lower rate of scala translocation [16,17]. Herein, we report our first experience of rob-EAI, with or without intraoperative visual feedback through fluoroscopy. To our knowledge, this intraoperative procedure associating fluoroscopy feedback together with rob-EAI has never been reported to date. We reveal the advantages and pitfalls that can be encountered when using this combination of rob-EAI with fluoroscopy feedback.

## 2. Materials and Methods

### 2.1. Chart Inclusion

From November 2021 to October 2022, all the candidate adults for a cochlear implantation were informed that a rob-EAI would be attempted, with or without fluoroscopy, depending on the availability of the hybrid operating room equipped with a robotic cone-beam C-arm system (Angio suite, Imabloc). This hybrid operating room is located in the radiology department of our institution and can host several types of procedures, e.g., cardiovascular, gynecological, and oncological procedures. This Angio suite is currently available for us once every two weeks. In all cases, preoperative imaging of the head and petrous bone was routinely prescribed but not all patients underwent this specific imaging in our institution. Because both rob-EAI and visual control of EA-insertion through fluoroscopy are routine procedures for cochlear implantation in our institution, no additional administrative procedure apart from providing clear information was required. All the patients were free to refuse the use of their personal data for scientific purposes, but none did so. This non-refusal procedure, consisting of the retrospective analysis of patients’ records, was subject to the approval of our local ethics committee (00013412, “CHU de Clermont-Ferrand IRB #1”, IRB number 2022 CF-079) in compliance with the French policy of individual data protection.

### 2.2. Surgical Procedure

#### 2.2.1. In the Regular Operating Room of Our Otolaryngologic Department

The patient was installed on a surgical bed with a flat head support, bordered by metal rails. This is mandatory for securing the command of the robot (rob-space mouse; rob-SM) to the upper rail, just up to the vertex of the patient. The head was securely placed on a circular head support, with a frontal adhesive tape, attached to the table in the relevant position, i.e., turned to the opposite site of the ear being operated on, with a slight extension of approximatively 15–20°. In addition, to prevent any slipping of the patient’s body when rotating the table, two straps were placed, one at the chest, the second at the lap. The table was then rotated 12° forward, and 5° backward before surgery, to check the stability of the patient’s head on the table. Once the patient had been installed, the surgical team, consisting of a surgeon, a resident doctor and an instrument nurse, could prepare the operative field and drape both the microscope and the robot.

The surgical approach to the cochlea was a regular transmastoid route with posterior tympanotomy, achieved through microscopic control. Once the round window was clearly exposed, the cochlear implant was placed in a subperiosteal area with or without drilling a dedicated bed in the temporal bone. Then, the tip of the EA was delicately positioned at the round window, after the round window membrane had been opened.

The robot used was the RobOtol^®^ (Collin Medical, Paris, France). This device is equipped with a robotized arm (RA) capable of moving with six degrees of freedom. The RA movements were controlled by the manually actuated rob-SM. It is important to note that this rob-SM allows RA movements in several directions but can also be locked in translation only (in the three spatial dimensions), or in rotation only. It is also possible, once an adequate axis is chosen, to fix the axis so that translation only along this axis is possible. Three types of RA movement speeds can be chosen: high speed, regular, or slow speed. When choosing slow-speed movements, insertion is never faster than 1 mm/s. It can be slowed by adjusting the hand pressure on the rob-SM button. The specific cochlear implant holder (CIHo) is unique for each brand of cochlear implant. During the surgical procedure, the RobOtol^®^ is directed towards the head of the patient, in front of the surgeon, perpendicular to the table, and secured in the position chosen (Figure 1).

The RA was then moved above the head so that the CIHo could be directed towards the surgical route to the round window. The cochlear implant EA was then secured to the CIHo, either by inserting the ailette in the specific fork-shaped tip of the CIHo for CI622^®^ cochlear implants (Cochlear, Sydney, Australia), or by placing it in a specific gutter at the tip of the CIHo for the two other brands used, i.e., Flex 24^®^ or Flex 26 ^®^ (Med El, Innsbruck, Austria) and Neuro ZTI CLA or EVO (Oticon Medical, Vallauris, France). Finally, the EA could be inserted, under microscopic control, using the rob-SM at low speed, at less than 1 mm/s. Once the insertion had been achieved, the EA was detached from the CIHo, then the RA withdrawn from the mastoid, and the RobOtol^®^ moved back from the operating field. The EA could then be rolled down within the mastoid cavity, the ground electrode, if any (depending on the brand of cochlear implant) placed in a temporal pocket muscle, after which the wound was closed layer by layer. Finally, the patient underwent a postoperative petrous bone computed tomography (CT)-scan before being discharged on the same, or on the following, day.

#### 2.2.2. Surgical Procedure in the Angio Suite Imabloc

The patient was installed in the same way as in our regular operating room. However, the Robotol^®^ and the robotized C-arm of the cone beam (Artis Zeego, Siemens, Erlangen, Germany) had to be positioned in the expected surgical position to check that the radiological robotized C-arm could be moved without touching the microscope or the RobOtol^®^. Sufficient room and careful placement are required to ensure feedback from the rob-EAI under microscope with fluoroscopy (Figure 2).

The following rob-EAI did not differ from the regular operating room, except that the surgeon took several minutes to put on a protective leaded jacket before switching on the fluoroscopy. In some cases, the surgeon could leave the hybrid room, and perform a remote EAI, with visual control of the surgical field recorded from the microscope camera together with the fluoroscopy (Figure 3).

Surgical and fluoroscopy views were projected on the same large screen facing the surgeon outside the hybrid room, protected by an anti-radiation screen (Figure 4).

### 2.3. Postoperative Imaging

Patients operated in our regular operating room underwent a postoperative CT-scan of their petrous bones without contrast the same or the following day. For those operated in the Angio suite Imabloc, postoperative 3D cone beam acquisition was carried out on the operating table before the patient was awakened.

### 2.4. Data Collection

Patient characteristics were recorded anonymously. The additional time of installation due to the RobOtol^®^, and to the checking of the double installation of the RobOtol^®^ and the robotized C-arm fluoroscope when using the hybrid room was recorded. The duration of RobOtol^®^ use during the surgical procedure was also recorded, depending on which procedure was used, i.e., regular operative room (ROR), Angio suite Imabloc (Imabloc), or Angio suite Imabloc with remote EA-insertion (Remote-EAI). The success or failure of EA-insertion with the RobOtol^®^ was recorded. The angle of insertion was collected for each patient. Any misrouting of the EA was noted. Particular attention was given to cases with failure of the rob-EAI to identify its cause. The postoperative petrous bone cone beams, or the postoperative petrous bone CT-scans, were analyzed by the radiology teams to determine if any translocation of the EA was suspected. When the basilar membrane was identified on the preoperative magnetic resonance imaging (MRI), the postoperative petrous bone cone beam, or the postoperative petrous bone CT-scan, was merged with the pre-operative three dimentional (3D) T2 space MRI by the radiology team to identify translocation. (coregistration fusion with advantage Workstation (ADW) 4.5, General Electric). When the basilar membrane was not identified on the preoperative MRI, only the postoperative petrous bone cone beams or the postoperative petrous bone CT-scans were analyzed, based on the aspect and location of the EA in maximum intensity projection (MIP), to determine if any translocation of the EA was suspected.

### 2.5. Statistics

The comparisons used an ANOVA and a post hoc Scheffé test. A value for *p* ≤ 0.05 was considered significant.

## 3. Results

Table 1 summarizes the patients’ characteristics and indicates whether the robotized EA insertion was successful. Among the 23 patients (10 females, 61+/−19 y.o.) 19 (82.6%) were implanted successfully with a rob-EAI. The rob-EAI with the RobOtol^®^ required on 18+/−7 min average for the entire series.

### 3.1. Patients Operated on in the Angio Suite Imabloc

Eleven cases were implanted under fluoroscopy feedback in the Angio suite Imabloc associated with the implementation of rob-EAI. Among all the patients implanted in the Angio suite, remote-EAI was attempted in eight cases (72.7%). Regarding the three cases where the robotized EAI failed, they were eventually successfully implanted by hand using a classical procedure. In these three cases, failure was due only to our lack of experience with this double control of EA-insertion. Patient 9 failed because the tip of the EA was not securely positioned at the round window before remote-EAI. Therefore, the EA went out of the round window at the very beginning of the remote EAI. The EA bent in a spring shape out of the cochlea. It was slowly withdrawn, manually realigned, then slowly inserted by hand. The operation on patient 5 failed because the table changed its position every time the assistant radiologist tried to position the robotized C-arm, which was not compatible with the positioning of the CIHo required. The operation on patient 23 failed because the surgeon focused on the fluoroscopy scene instead of the surgical field, therefore they did not detect the misrouting in the mastoid of the CIHo, only the late EA- bending on the fluoroscopy screen. The EA was removed, manually realigned, and then slowly inserted by hand. In one case (patient 2) the EA inserted showed an initial square shape on the fluoroscopy screen, with a limited angle of insertion at 360°. The EA was thus removed using the RobOtol^®^ after which a second rob-EAI with a suitable round shape was inserted successfully, allowing for a full rob-EAI with a 540° angle. Nobody could have detected that the initial insertion was square-shaped if it had not been displayed on the fluoroscopy scene. Regarding the Imabloc patients, the time dedicated to the robot was 23+/−7 min, (22+/−9 min in Remote-EAI patients; 18+/−3 min with no remote EA-insertion, the difference was not significant, *p* = 0.65).

### 3.2. Patients Operated in the Regular Operated Room

#### 3.2.1. Global Results

Among these 22 cases, 21 (95.4%) underwent a successful rob-EAI. The only case of failure of rob- EAI occurred in patient 17, where an extensive fibrosis of the basal turn almost precluded cochlear implantation. In this case, the deep cochlear fibrosis made smooth EAI impossible. After having clearly identified the obstruction of the entire basal turn by fibrosis, we decided to largely open the basal turn to allow the correct removal of the fibrosis. We were obliged to force the residual fibrosis obstructing the end of the basal turn with a rigid dummy EA before achieving a difficult insertion of the EA by hand. In this case the rob-EAI was of no use. The additional time due to the use of the RobOtol^®^ was on average 15+/−4 min. Although shorter than in the Angio suite Imabloc, this duration did not significantly differ, neither with the group implanted under fluoroscopy in the Angio suite Imabloc (*p* = 0.79), nor with the remote-EAI group; *p* = 0.10).

#### 3.2.2. Quality of rob-EAI

The angle of insertion was on average 477°+/−63 among the 22 patients with no attempt at hearing preservation. All the patients had a full insertion except one with a Mondini malformation (type 2 incomplete cochlear partition). In the sole patient (patient 19) whose hearing was tentatively preserved, the desired angle at 360° was obtained. Hearing was preserved, with a mean postoperative pure tone average (PTA)between 250 Hz–4 kHz at 88 dB versus 83 dB in the preoperative period, which was not significant. The patient spontaneously reported that she could still hear her intrinsic throat noises the same day after surgery.

Whatever the type of rob-EAI, we never succeeded in defining a unique perfect axis leading to a full insertion. In all the patients, despite having determined a likely adequate preinsertion axis of the CIHo, we had to correct it during insertion, using the rob-SM to adjust the EA trajectory.

Postoperative CT scans and preoperative MRIs of the cochlea made it possible for the radiologist to merge postoperative CT and preoperative MRI images when the basilar membrane was identified on MRI. In three cases the resolution of MRI and CT scans were sufficient to build these merged images (Figure 5).

It was clear that the EA was not translocated in these three cases. We did not suspect any scala translocation in the other cases, based on the comparisons of their postoperative CT-scans with the three cases with merged cochlear images.

## 4. Discussion

This preliminary report demonstrates for the first time that cochlear implantation with rob-EAI can be coupled with visual feedback through fluoroscopy. It confirms that a rob-EAI is feasible in most patients. To our knowledge, this is the first time that such remote EAI has been reported in cochlear implantation. To date, several teams has reported this type of EA-insertion using RobOtol^®^ but none of them used fluoroscopy as visual feedback [7,17,18]. Recently, a cochlear implantation under fluoroscopy with the HEARO procedure has been reported [19]. In this interesting report, while the approach was robotized, the EA insertion was not, which is quite different from our procedure. In this series, we did not use intraoperative electrocochleography (EcoG). It is interesting to note that intraoperatively EcoG during a rob-EAI using the RobOtol^®^ has been proven feasible [18]. In order to improve the chance of hearing preservation, a rob-EAI could be advantageously associated with fluoroscopy and intraoperative EcoG monitoring. This combination of different tools for a smooth and mini-invasive EAI needs further studies to highlight its interest in preserving hearing. The use of a robotized EA-insertion leads to steady and low-speed insertion that has been proven to be less deleterious to the cochlea [7,17,20]. Therefore, rob -EAI could be the preferred technique of cochlear implantation if it is capable of harmonious and steady EA-insertion at a low speed. Here, we revealed the additional advantage of using fluoroscopy, which provides visual feedback. Case 2 was particularly demonstrative of this advantage, for which the first robotized EA-insertion ended in a square-shaped insertion with a limited angle at 360° that was detected by fluoroscopy and then corrected. It should be mentioned that the RobOtol^®^ provides no haptic feedback. We know that even with the haptic feedback of the human hand, in some cases it is very difficult to feel certain errors during EA-insertion, such as EA-tip-foldover or EA-misrouting. Hopefully, these errors can be revealed by fluoroscopy [15]. It is therefore not surprising that, with no haptic control at all, a rob-EAI can result in a divergent EA winding around the modiolus. Thus, case 2 convinced us to use fluoroscopy feedback with the RobOtol^®^ as much as possible. Another positive point of using a hybrid room like the Angio suite Imabloc is that the RobOtol^®^ permits remote-EAI under fluoroscopy, limiting the irradiation delivered to the operating staff. The total fluoroscopic time has been previously precisely reported [15]. Because fluoroscopy only served for control of the EA- insertion, the exposition time of patients to radiation was very low, less than 5 min. Herein, the use of the RObOtol^®^ did not increase the time of fluoroscopy. The EA-insertion was always performed at low speed (<1 mm/s) even before we used the RobOtol^®^. This time was reported previously at less than 4 min [15]. The total radiation dose was calculated at 4053+/−1994 μGray m^2^, including the radiation due to the final cone beam CT scan responsible for the most part of the X-ray dose. From a clinical standpoint, it is equivalent to less than four digital substrate radiographs (DSA), which is acceptable in adults. In the case of children, we had recommended to skip the final cone beam CT scan, avoiding 96% of the total radiation dose, which was thus responsible for a low irradiation, less than one DSA [15]. By contrast, the total radiation dose can become harmful to the surgeon who repeatedly performs operations, as their hands are exposed to the radiation of fluoroscopy. Alternative means of protecting the surgeon’s hands from radiation do not as yet exist. While special anti-radiation gloves seem to protect against indirect scattered radiation [21], paradoxically, they could increase the dose of radiation received by direct exposure [22]. In addition, they are thicker than regular surgical gloves, which is uncomfortable for the delicate handling of micro-instruments. As soon as a remote EAI proved possible, we therefore systematically used it. However, among the eight patients who underwent remote-EAI, two cases failed. We now know, after patients 17 and 23, that when attempting a remote- EAI, it is necessary to double check that the EA tip is well secured at the entrance of the round window and focus on the surgical field rather than on the fluoroscopy scene.

For us, the additional time required to use the RobOtol^®^ in the Angio suite Imabloc was acceptable at around 23 min, vs. 15 min in the regular operation room. This double additional refinement of EAI, i.e., rob-EAI with visual fluoroscopy feedback, which ensures optimal EAI in selected cases, makes it worthwhile to spend a bit less than 30 additional minutes in the operating room. The postoperative control of the EA- scala position, was achieved here in three cases by merging preoperative MRIs and postoperative CT-scans. When these images are of high quality, the technique appears to be very accurate, as it allows the direct visualization of the basilar membrane, in contrast to other techniques based only on CT-scans [23]. However, this merging technique requires high quality images. Today we systematically ask for high quality preoperative labyrinthine MRIs and postoperative CT-scans, in order to build-up merged pre and postoperative images of the cochlea for all patients. Thus, it is likely that no translocation occurred in our series with rob-EAI.

In the case of basal fibrosis, such as in case 17, robotized-EAI becomes very hard to achieve. The lack of haptic control makes it difficult to force the obstacle with a suitable force of insertion while avoiding any EA bending. In this situation, the haptic feedback of an experienced surgeon is much preferable and can be helped by fluoroscopy [15]. Also, because the basal fibrosis has to be partially removed and the EA forced through the fibrosis, the objective is no longer to perform a minimally invasive insertion, but to achieve it.

It must be born in mind that the RobOtol^®^ is currently well-suited for straight EAs, but for only one perimodiolar EA (Mid-scala from Advanced Bionics^®^). This is a real limitation in cases where the cochlear implant team would prefer such EAs, for instance in certain cases of otosclerosis to minimize the current diffusion through the diseased bone [24]. Herein, we did not select patients with the need of perimodiolar EAI, as the Mid-scala EA from Advanced Bionics^®^ is not currently available in our institution.

## 5. Conclusions

In conclusion, our preliminary cochlear implant cohort using the RobOtol^®^ confirmed that rob-EAI is feasible in almost all cases, except those with extensive cochlear fibrosis. In addition, combining the use of a rob- EAI and fluoroscopy feedback, which permits remote EAI, can be done with safety for both the patient and the surgeon in most cases.

## Figures and Tables

**Figure 1 jcm-12-00211-f001:**
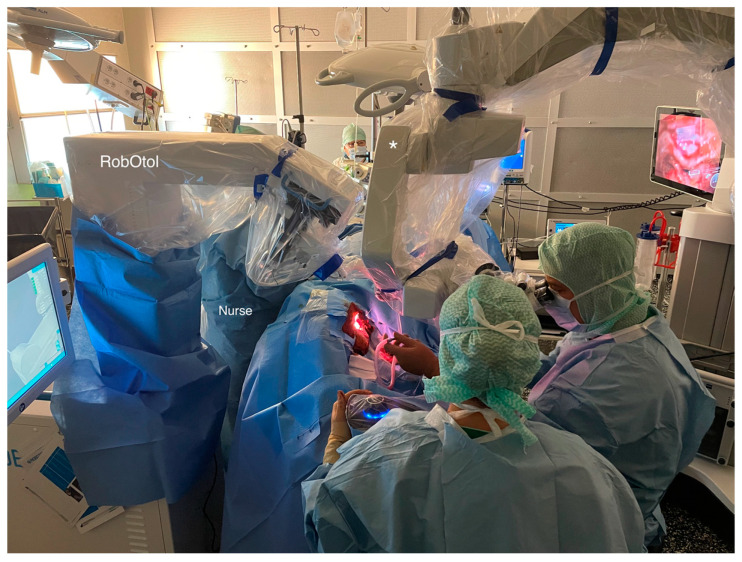
Intraoperative view showing the placement of the RobOtol^®^ and microscope in the regular operating room (* marks the operating microscope).

**Figure 2 jcm-12-00211-f002:**
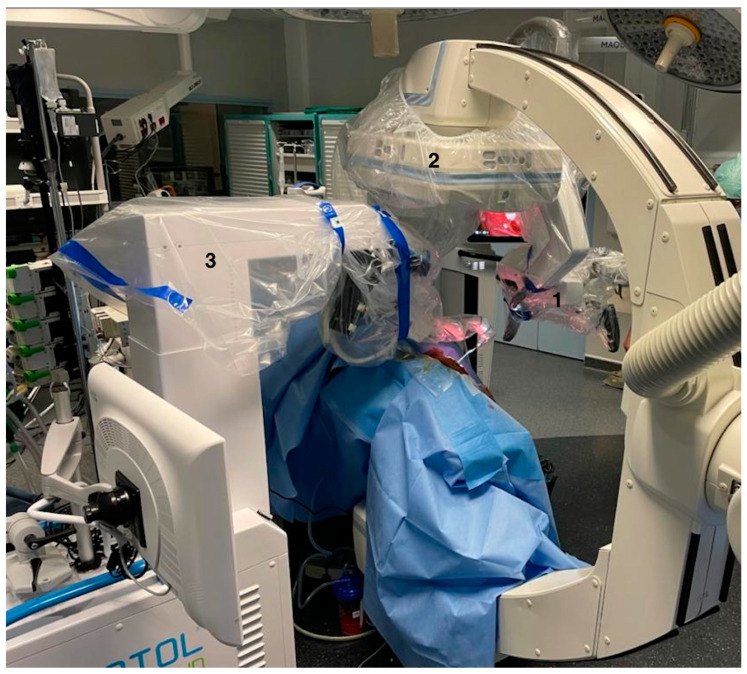
Intraoperative view showing the placement of the RObOtol^®^ (3), microscope (1) and radiological robotized C-arm (2).

**Figure 3 jcm-12-00211-f003:**
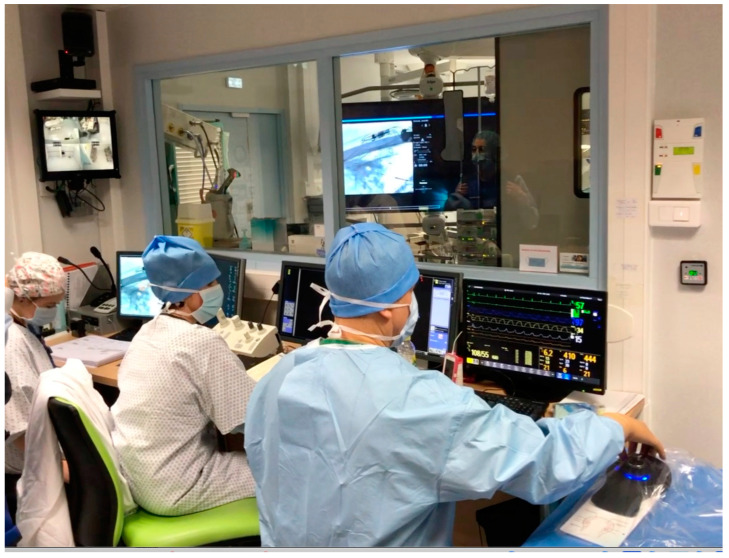
The surgeon can start the rob- EAI remotely outside of the Angio suite, sitting next to the anesthetists.

**Figure 4 jcm-12-00211-f004:**
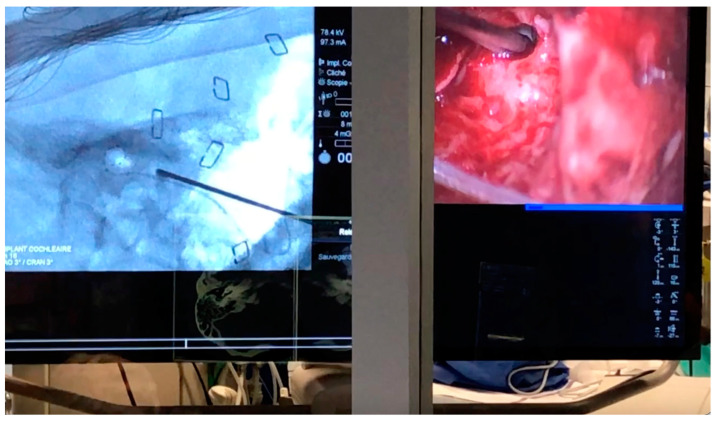
Simultaneous view of the surgical and fluoroscopy screen, as seen behind the protective shield at the end of a remote-EAI.

**Figure 5 jcm-12-00211-f005:**
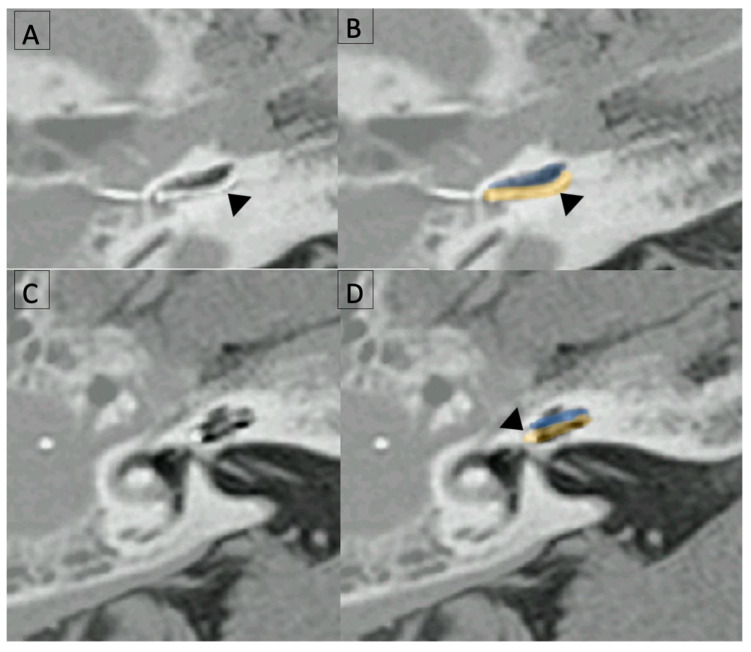
Images from the same patient (M8) obtained by merging preoperative MRI and postoperative CT-scans. Black arrow heads point to the EA. (**A**) postoperative CT-scan axial slice of the basal turn; (**B**) Merged images at the same level as (**A**) after identification of the scala tympani (yellow) and vestibuli (blue) on MRI; (**C**) postoperative CT-scan axial slice of the second turn; and (**D**) Merged images at the same level as (**C**) after identification of the scala tympani (yellow) and vestibuli (blue) on MRI. In this example, EA was fully inserted into the scala tympani.

**Table 1 jcm-12-00211-t001:** Patients’ characteristics.

Patient	Sex	Age (Years)	Cochlear Implant Brand	Imabloc	Time of Robot Use (Min)	Remote EAI	Success
1	M	60	COCHLEAR CI622	no	13	no	YES
2	M	44	COCHLEAR CI622	yes	20	no	YES
3	M	83	MED-EL FLEX 24	no	18	no	YES
4	F	76	MED EL FORM 24	yes	18	YES	YES
5	F	72	COCHLEAR CI622	yes	20	no	no
6	M	79	COCHLEAR CI622	no	18	no	YES
7	M	82	COCHLEAR CI622	yes	22	YES	YES
8	F	60	COCHLEAR CI622	no	18	no	YES
9	M	57	COCHLEAR CI622	yes	19	YES	no
10	M	77	COCHLEAR CI622	Yes	25	YES	YES
11	M	74	COCHLEAR CI622	yes	21	no	YES
12	M	37	COCHLEAR CI622	yes	40	YES	YES
13	M	70	COCHLEAR CI622	no	25	no	YES
14	F	40	COCHLEAR CI622	no	8	no	YES
15	M	79	MED EL FLEX 24	no	14	no	YES
16	M	15	COCHLEAR CI622	yes	23	YES	YES
17	M	26	COCHLEAR CI622	no	N/A	no	no
18	F	66	MED EL FLEX 26	no	12	no	YES
19	F	42	COCHLEAR CI622	no	13	no	YES
20	F	74	COCHLEAR CI622	yes	15	no	YES
21	F	73	COCHLEAR CI622	yes	23	YES	YES
22	F	57	OTICON MEDICAL NZTI CLA	no	12	no	YES
23	F	54	COCHLEAR CI622	yes	9,45	YES	no

## Data Availability

All data supporting our results are included in the text, table, and figures.

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
