# Peer review of "Robotized Cochlear Implantation under Fluoroscopy: A Preliminary Series"

_jcm, 2022, doi:10.3390/jcm12010211_

Round 1
Reviewer 1 Report
The authors describe for the first time a fluroscopy evaluation during a ROBOTOLguided CI electrode insertion. They even performed this additionally in some cases as a remote procedure out of the direct surgical theatre room. This is an additionally challenging step into the complete online performance of an electrode insertion. The study is well structured and in general fine. Two things needs to be improved. The authors should handle extensively the limitations and must discuss the fixation of the head. Uncontrolled head movements could have a desastrous effect and this problem must be solved (e.g. Layla etc...). The second point is the English. The whole manuscript must be checked by a native speaker.
Author Response
Please see the attacment

Reviewer 2 Report
Nowadays the advances in cochlear implantation have focused on minimizing cochlear trauma to improve hearing preservation outcomes, and in doing so expanding candidacy to patients with useful cochlear reserve. At the same time recent advancements in robotic surgery have set a growing body of evidence for the clinical application of the robotized cochlear implantation with many potential benefits. This approach holds promise as a potential tool to minimize intracochlear trauma with electrode insertion, improve surgical efficiency, and reduce surgical complications. At the same time the fluoroscopy gave authors the possibility to confirm the right scala placement and absence of translocation.
Authors evaluated the use of the RobOtol® (Collin, Bagneux, France) otologic robot to insert CI electrodes into the inner ear with or without intraoperative visual feedback through fluoroscopy. A robotized EA insertion (rob-EAI) has been shown to allow for less traumatic insertion, with a lower rate of scala translocation. This intraoperative procedure associating fluoroscopy feedback together with rob-EAI has never been reported to date.
For the first time it was demonstrated that cochlear implantation with rob-EAI can be coupled with visual feedback through fluoroscopy. It was confirmed that a rob-EAI is feasible in most patients and this is the first time that such remote EAI is reported in cochlear implantation. The use of a robotized EA-insertion leads to a steady and low-speed insertion that has been proven to be less deleterious to the cochlea. It was concluded that the rob -EAI thus could be the preferred technique of cochlear implantation if it could lead to a harmonious and steady EA-insertion at low speed. The additional advantage of using fluoroscopy which provides visual feedback was found.
In conclusion, the preliminary cochlear implant cohort using the RobOtol® confirms that rob-EAI is feasible in almost all cases, but those with extensive cochlear fibrosis. In addition, combining the use of a rob- EAI and fluoroscopy feedback, allowing a remote EAI and can be done with safety both for patient and surgeon in most cases.
Advancements in cochlear implant surgical approaches and electrode designs have enabled preservation of residual acoustic hearing. Preservation of low-frequency hearing allows cochlear implant users to benefit from electroacoustic stimulation, which improves performance in complex listening situations, such as music appreciation and speech understanding in noise. Despite the relative high rates of success of hearing preservation, postoperative acoustic hearing outcomes remain unpredictable.
Authors used only straight cochlear electrode arrays - CI622® (Cochlear, Sydney Australia), Flex 24® or Flex 26 ® (Med El, Innsruck, Austria) and Neuro ZTI CLA or EVO (Oticon Medical, Vallauris, France).
Question – are any limitations the use with suggested technique of perimodiolar electrode arrays? It would be nice to cover this topic as well.
To conclude, the article submitted opens new research capabilities in patients with cochlear implants and residual hearing, widening the existing theoretical and practical knowledge regarding robotic CI surgery combined with fluoroscopy and further research progress in this area
